# Annealing Machine-assisted Learning of Graph Neural Network for Combinatorial Optimization

**Pablo Loyola**[1]     **Kento Hasegawa**[3]     **Andres Hoyos-Idobro**[2]     **Kazuo Ono**[3]
**Toyotaro Suzumura**[1,4]     **Yu Hirate**[1]     **Masanao Yamaoka**[3]

[1]Rakuten Institute of Technology, Rakuten Group, Inc., Tokyo, Japan
[2]Rakuten Institute of Technology, Rakuten Group, Inc., Paris, France
[3]Hitachi, Ltd., Tokyo, Japan
[4]The University of Tokyo, Japan
`{pablo.a.loyola, andres.hoyosidrobo, yu.hirate}@rakuten.com`
`{kento.hasegawa.bc, kazuo.ono.ap, masanao.yamaoka.ns}@hitachi.com`
`suzumura@acm.org`

## Abstract

While Annealing Machines (AM) have shown increasing capabilities in solving complex combinatorial problems, positioning themselves as a more immediate alternative to the expected advances of future fully quantum solutions, there are still scaling limitations. In parallel, Graph Neural Networks (GNN) have been recently adapted to solve combinatorial problems, showing competitive results and potentially high scalability due to their distributed nature. We propose a merging approach that aims at retaining both the accuracy exhibited by AMs and the representational flexibility and scalability of GNNs. Our model considers a compression step, followed by a supervised interaction where partial solutions obtained from the AM are used to guide local GNNs from where node feature representations are obtained and combined to initialize an additional GNN-based solver that handles the original graph's target problem. Intuitively, the AM can solve the combinatorial problem indirectly by *infusing* its knowledge into the GNN. Experiments on canonical optimization problems show that the idea is feasible, effectively allowing the AM to solve size problems beyond its original limits.

## 1  Introduction

Graph-based approaches are one of the most predominant techniques for learning combinatorial optimization solvers. Their distributed nature allows them to scale up to millions of nodes [33, 19]. Nevertheless, their probabilistic nature, which provides soft assignments to decision variables, may produce solutions at a different level than their classic counterparts [25]. Annealing Machines (AM), a concurrent line of research, is seen as a more immediate alternative to the expected advances of future fully quantum solutions, which are currently in use in several industries [30]. Their major drawback, similarly suffered by the fully quantum versions, is their scalability, handling a limited number of variables, sometimes forcing problem reformulations to fit hardware limitations [28].

We see these two approaches working towards the same goal: *i)* GNN-based methods enable scalability, but their solutions could be noisy; *ii)* AM-based methods provide high precision, yet they are limited in the number of variables they can handle. This apparent trade-off motivates us to design a framework to capture the best of each technology: high scalability and high precision. This work proposes a way to *combine* the solving capabilities of both graph and annealing-based methods into a single workflow that handles combinatorial optimization problems at scale.

Second Workshop on Machine Learning with New Compute Paradigms at NeurIPS 2024(MLNCP 2024).

Given a combinatorial problem $P$ and its associated graph $\mathcal{G}_P = (\mathcal{V}_P, \mathcal{E}_P)$, with node set $\mathcal{V}_P$ and edge set $\mathcal{E}_P$, our framework considers a sequential compression step, that generates a list $\{\mathcal{G}_i\}_{i=1}^s$ of compressed versions of $\mathcal{G}_P$ with $|\mathcal{V}_P| \geq |\mathcal{V}_1| \geq \ldots \geq |\mathcal{V}_s|$. Then, we perform a supervised interaction step where the AM solution for each graph $\mathcal{G}_i$ acts like target labels of a $i$-th local GNN, $\forall i \in [s]$. Finally, we pool all these node feature representations and use them to initialize an extra GNN-based solver that handles $P$ on the original graph $\mathcal{G}$. In that sense, the AM solves the combinatorial problem indirectly by infusing its knowledge into the GNN. We conducted an empirical study on three canonical optimization problems over various families of graphs. Our results show that the proposed solution is feasible, allowing the AM to solve problems of size beyond its initial scope, reduce the overall converge time, and, in some cases, even increase the solution quality.

## 2 Background and Related Work

**Annealing Machines for Combinatorial Optimization** AM can operate based on various mechanisms, including both quantum and classical: superconducting flux qubits, degenerating optical parametric oscillators, and semiconductor CMOS integrated circuits. However, there are various types of AM since D-Wave released the first commercial quantum AM [18, 31, 23, 30, 12, 11, 14, 17, 1, 20]. AM requires casting the target problem as a Quadratic Unconstrained Binary Optimization (QUBO) problem [10, 8]. The QUBO formulation models the problem as a graph, with nodes as decision variables and edges as (energy) couplings that encode their relationships. This formulation has proven versatile, with AM covering a wide range of applications [7, 26, 6, 22, 24, 21, 16, 32]. We refer the reader to [10] for a comprehensive QUBO formulation tutorial. We consider the following problem:

$$
\begin{aligned}
&\text{minimize } \mathbf{x}^\mathsf{T} \mathbf{Q} \mathbf{x} =: H_{\mathrm{QUBO}}(\mathbf{x}) \\
&\text{subject to } \mathbf{x} \in \{0, 1\}^n,
\end{aligned}
\tag{1}
$$

where $H_{\mathrm{QUBO}}$ is the Hamiltonian associated with the QUBO matrix $\mathbf{Q} \in \mathbb{R}^{n \times n}$, which is the symmetric matrix that encodes the target problem, and $\mathbf{x}$ is the vector of binary assignments. In this work, we explore to what extent the GNNs can enable AMs to handle larger problems, hopefully without sacrificing accuracy.

**QUBO-based Graph Learning for Combinatorial Optimization** Recently [25] exploited the relationship between GNNs and QUBO for solving combinatorial problems. This method relies on the QUBO formulation of a target problem, where a GNN takes the Hamiltonian as the cost function and minimizes it in an unsupervised way. Thus, for the $k$-th GNN layer and a given node $\mathbf{v} \in \mathcal{V}$, we obtain a node feature representation that depends on both of its previous representation at the $(k-1)$-th layer and the aggregated representations of the direct neighbors,

$$
\mathbf{h}_v^k = \Phi_\theta \left( \mathbf{h}_v^{k-1}, \left\{ \mathbf{h}_u^{k-1} \middle| \ \forall u \in \mathcal{N}_v \right\} \right), \forall k \in [K],
\tag{2}
$$

where $\mathbf{h}_v^k \in \mathbb{R}^{d_k}$, $d_k$ is the dimension of the $k$-th representation, $\mathcal{N}_v$ is the set of neighbors of $v$, and $\Phi_\theta$ a learnable function [15]. The resulting representation passes through a linear layer and an activation function to obtain a single positive real value, and is then projected to integer values to obtain a final binary node assignment.

We write it in compact form as:

$$
\mathbf{x}^{\mathrm{GNN}} = \psi_\omega(\underbrace{\mathrm{GNN}_\theta(\mathcal{G}, \mathbf{F})}_{= \bar{\mathbf{F}} \in \mathbb{R}^{|\mathcal{V}| \times d_K}}),
\tag{3}
$$

where $\mathbf{F} \in \mathbb{R}^{|\mathcal{V}| \times d_0}$ is the matrix of initial node features[1], $\mathrm{GNN}_\theta$ maps the graph $\mathcal{G}$ and $\mathbf{F}$ to $\bar{\mathbf{F}}$, and $\psi_\omega : \mathbb{R} \to \{0, 1\}$ is the composition of a linear layer and an activation function. Let $\mathbf{f}_v \in \mathbb{R}^{d_0}$ be the $v$-row of $\mathbf{F}$, the feature vector of the node $v$. Thus, the initial embedding in Eq. 2 corresponds to $\mathbf{h}_v^0 = \mathbf{f}_v, \ \forall v \in \mathcal{V}$. Therefore, finding a solution $\mathbf{x}^{\mathrm{GNN}}$ amounts to minimizing $H_{\mathrm{QUBO}}(\mathbf{x}^{\mathrm{GNN}})$ in an unsupervised manner.

---

[1]We assume that the ordering of the nodes is consistent with the ordering in the adjacency matrix.

# 3 Methodology

**Problem Statement** Our main goal is to assess AM and GNN complementarity. We assume a scenario where the target problem is large enough that it cannot be solved solely by the AM. Therefore, we propose a framework that divides the problem into smaller pieces so the AM can consume and solve, and the GNN can act as a bridge, aggregating information to achieve a global solution.

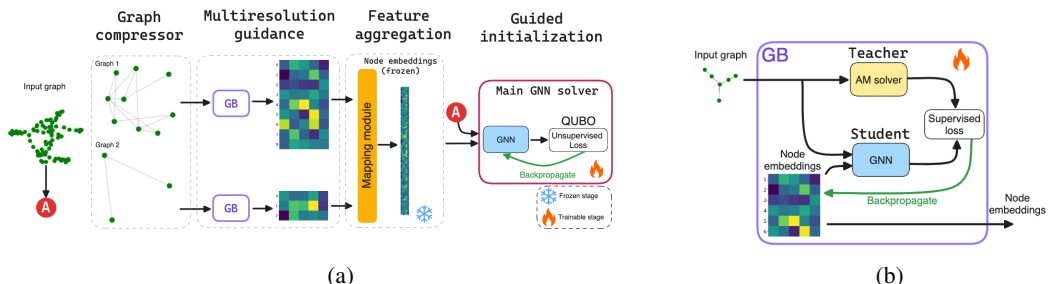

(a)  (b)

Figure 1: (a) Proposed approach: Original graph $\mathcal{G}_P$ is compressed into a sequence of decreasing size problem graphs, which are solved by AM. A local GNN solver uses those solutions as guidance, and their resulting learned node representations are aggregated through a *Mapping module* to initialize a Main GNN solver that attaches to the original graph $\mathcal{G}_P$. (b) Guiding block (GB) diagram.

**Graph Compression** We reduce the size of the original graph $\mathcal{G}_P$ using Louvain decomposition for network community detection [2]. The output of this decomposition is a list of size-decreasing graphs $\{\mathcal{G}_i\}_{i=1}^s$. For each $\mathcal{G}_i = (\mathcal{V}_i, \mathcal{E}_i)$, the algorithm admits a mapping back to the original graph. Thus, we can relate a single artificial node $n \in \mathcal{G}_i$ with the corresponding set of actual nodes in the original graph $\mathcal{G}_P$. We ensure the size of all resulting graphs is smaller than the aforesaid feasibility limit exhibited by the AM. We chose Louvain decomposition because *i)* it is one of the most cited and well-understood methods for community detection *ii)* it is hierarchical, allowing reconstruction from a given granularity to the original graph, *iii)* among hierarchical models, Louvain provides the most homogeneous results, as literature shows [9]. Given the expected diversity of graphs our method should handle, we consider homogeneity to be a desirable factor.

**Multiresolution Guidance I : Locally-assisted Solvers Interaction** We get a QUBO matrix $\mathbf{Q}_i$ for each $\mathcal{G}_i$. We assume that while these derived matrices are not equivalent to $\mathcal{G}_P$, there should be certain alignment as they are working on different granularities of the original graph $\mathcal{G}_P$. As each $\mathcal{G}_i$ is smaller than the AM's limit, we can apply it to solve them. This step outputs, for each $\mathcal{G}_i$, the solution found by the AM, in the form of a binary solution vector $\mathbf{x}_i^{\mathrm{AM}} \in \mathbb{R}^{|\mathcal{V}_i|}$.

**Multiresolution Guidance II : Guiding block (GB)** We consider AM solutions a good source of supervision and use them to drive local GNNs that solve upon the same set of $\mathcal{G}_i$ graphs. Instead of just minimizing the Hamiltonian in an unsupervised way, as described in Sec. 2, we propose to combine the Hamiltonian cost with a measure of alignment between the AM's solution and GNN's partial solutions at each training time/epoch $t$. For a graph $\mathcal{G}_i$, let $\mathbf{x}_{i,t}^{\mathrm{GNN}}$ be the GNN's partial solution. $\mathbf{x}_{i,t}^{\mathrm{GNN}}$ denotes assignment scores. Thus, we have $\theta^i = \arg\min_{\theta \in \Theta} \ell\left(\mathbf{x}_{i,t}^{\mathrm{GNN}}, \mathbf{x}_i^{\mathrm{AM}}\right) \ \forall i \in [s]$, where $\mathbf{x}_{i,t}^{\mathrm{GNN}} = \psi_\omega(\mathrm{GNN}_\theta(\mathcal{G}_i, \mathbf{F}))$. We set $\ell(\mathbf{a}, \mathbf{b}) = \|\mathbf{a} - \mathbf{b}\|^2$. *Local* GNNs can quickly converge for each compressed graph due to this guidance. While we check these local results' specific behavior and quality, we focus more on the final node representations. For each $\mathcal{G}_i$, this step gets a matrix $\bar{\mathbf{F}}^i \in \mathbb{R}^{|\mathcal{V}_i| \times d_K}$ (with $d_K$ a predefined vector dimensionality) with the node vectors after the GNN converged. Fig. 1b depicts this process for one compressed graph.

**Aggregating Partial Solutions** The output of the previous step gives a node feature vector for each compressed graph $\mathcal{G}_i$. We reuse these feature vectors to initialize a larger GNN to solve $P$ on the original graph $\mathcal{G}_P$, called *Main GNN solver* in Fig. 1a. We hypothesize that these feature representations associated with AM-guided solutions on compressed versions of $\mathcal{G}_P$ may encode valuable information that, if passed as initial node vectors for the main solver, could benefit the solution-finding process, in contrast, to initialize those node vectors randomly ($\mathbf{h}_v^0$ is random).

**Mapping Module** Let $\mathcal{V}_{n^i}$ denote the set of real nodes in the original graph $\mathcal{G}_P$ associated with a given node $n^i \in \mathcal{G}_i$, the $i$-th compressed version of $\mathcal{G}_P$. Thus, $\mathcal{V}_{n^i} = \{v \in \mathcal{V}_P \mid v \in \mathrm{Louvain}_{\mathrm{map}}(n^i)\}$,

where $\text{Louvain}_{\text{map}}$ is a lookup table (the Louvain mapping) that tracks back any artificial node to real nodes. We use the inverse degree as a normalization factor to *distribute* the learned feature representation associated with $n^i$. Formally, let $\deg : \mathcal{V}_P \to \mathbb{N}_0$ denote the degree function, then, given an artificial node $n^i \in \mathcal{G}_i$ with learned feature vector $\bar{\mathbf{f}}_{n^i} = \bar{\mathbf{F}}_n^i \in \mathbb{R}^{d_K}$, for each real node $v \in \mathcal{V}_{n^i}$ the associated feature vector will be $\bar{\mathbf{r}}_v^i = (\deg(v)/\sum_{u \in \mathcal{V}_{n^i}} \deg(u)) \, \bar{\mathbf{f}}_{n^i}$. Let $\bar{\mathbf{R}}^i \in \mathbb{R}^{|\mathcal{V}_P| \times d_K}$ be the matrix of synthetic node features, where each row of $\bar{\mathbf{R}}_v^i$ corresponds to $\bar{\mathbf{r}}_v^i$ for all $v \in \mathcal{V}_i$. These node features will be fixed from now on. We repeat the above process for all compressed versions $\mathcal{G}_i$ for $i \in [s]$, resulting in $s$ feature vectors for each real node $v \in \mathcal{V}$. Then, we obtain a single representation using any aggregation/pooling function. Our preliminary experiments showed that a simple average provided consistent results, $\mathbf{R} = \frac{1}{s} \sum_{i=1}^{s} \bar{\mathbf{R}}^i$.

**Guided initialization** At this point, we have obtained a feature vector associated with each node in the original graph, and we use them now to initialize a GNN-based solver that will be tasked to find a solution on the original graph $\mathcal{G}$. This step is simple: we construct a node features matrix $\mathbf{R} \in \mathbb{R}^{|\mathcal{V}_P| \times d_K}$ and use it as initialization for the GNN. Then, the solution procedure follows the standard unsupervised way described in Sec. 2. At the end of such a procedure, we finally obtain binary value vectors representing each node's solution assignments. Therefore, we minimize $H_{\text{QUBO}}(\mathbf{x}^{\text{GNN}})$, where $\mathbf{x}^{\text{GNN}} = \psi_\omega \left( \text{GNN}_\theta(\mathcal{G}_P, \mathbf{R}) \right)$. Algo. 1 summarizes this process.

---

**Algorithm 1** AM-assisted GNN solver

---

**Require:** graph $\mathcal{G}_P$, initial node features $\mathbf{F}$, learning rate of global solver $\alpha$, learning rate of local solver $\alpha_{\text{in}}$,

**Ensure:** $\mathbf{x}^{\text{GNN}} \in \underset{\mathbf{x} \in \{0,1\}^n}{\arg \min} \, H_{\text{QUBO}}(\mathbf{x})$

1: $\{\mathcal{G}_i\}_{i=1}^{s} \leftarrow \text{Louvain}(\mathcal{G}_P)$
2: **for** $i \in [s]$ **do**
3:     $\bar{\mathbf{F}}^i \leftarrow \text{GB}(\mathcal{G}_i, \mathbf{F}^i, \alpha_{\text{in}})$
4: **end for**
5: **for** $i \in [s]$ **do**
6:     **for** $v \in \mathcal{V}_i$ **do**
7:         $\bar{\mathbf{r}}_v^i \leftarrow \frac{\deg(v)}{\sum_{u \in \mathcal{V}_{n^i}} \deg(u)} \bar{\mathbf{f}}_{n^i}$
8:     **end for**
9: **end for**
10: $\mathbf{R} \leftarrow \frac{1}{s} \sum_{i=1}^{s} \bar{\mathbf{R}}^i$       ▷*Multiresolution aggregation*
11: **repeat**
12:     $\mathbf{x}^{\text{GNN}} \leftarrow \psi_\omega \left( \text{GNN}_\theta(\mathcal{G}_P, \mathbf{R}) \right)$   ▷*AM-guided GNN solution*
13:     $\theta \leftarrow \theta - \alpha \nabla_\theta H_{\text{QUBO}}(\mathbf{x}^{\text{GNN}})$   ▷*Update parameters*
14:     $\omega \leftarrow \omega - \alpha \nabla_\omega H_{\text{QUBO}}(\mathbf{x}^{\text{GNN}})$
15: **until** Convergence
16: **return** $\mathbf{x}^{\text{GNN}}$

---

# 4 Empirical Study

**Data Generation** We generated a set of synthetic random d-regular graphs of sizes $n \in [50\,000, 100\,000, 150\,000]$ and node degree $d \in [3, 4, 5]$, leading to a total of nine graphs. Random graphs are used for testing, as they allow us to evaluate the generalizability of our approach without the bias of structured adjacency matrices. The current AM has a limit of 100k nodes/variables, so the selected range allows us to study before and after such a feasibility threshold. For each graph, Louvain decomposition available on Networkx[13] was used.

**Optimization Problems** Let $\mathcal{V}$ and $\mathcal{E}$ be the set of all nodes and edges, respectively. *Maximum Cut (MaxCut)* finds a partition of nodes into two subsets such that the number of edges between different subsets is maximal. Its QUBO formulation is $H_{\text{MaxCut}}(\mathbf{x}) := \sum_{(i,j) \in \mathcal{E}} 2\mathbf{x}_i \mathbf{x}_j - \mathbf{x}_i - \mathbf{x}_j$, where $\mathbf{x}_i$ is 1 if node $i$ is in one set and 0 otherwise. *Maximum Independent Set (MIS)* finds the largest subset of nodes that are not connected. Its QUBO formulation of is $H_{\text{MIS}}(\mathbf{x}) := -\sum_{i \in \mathcal{V}} \mathbf{x}_i + \beta \sum_{(i,j) \in \mathcal{E}} \mathbf{x}_i \mathbf{x}_j$, where $\mathbf{x}_i$ is 1 if a node $i$ belongs to the independent set and 0 otherwise; $\beta > 0$ is the penalty coefficient. We used $\beta = 2$. *Graph Partition (GP)* partitions a graph into two equal-sized parts such that the number of divided edges is minimized. It is a generalization of MaxCut and its QUBO formulation is $H_{\text{GP}}(\mathbf{x}) := -H_{\text{MaxCut}}(\mathbf{x}) - \beta \sum_{i \in \mathcal{V}} ((1 - |\mathcal{V}|) \mathbf{x}_i + \sum_{j > i} 2\mathbf{x}_i \mathbf{x}_j)$. We used $\beta = 10$.

**Graph Neural Network** Both *Local* and *Main* GNN-based solvers have the same architecture: two convolutional layers (GraphConv) linked via ReLu activations. We pass the output from the second layer through a sigmoid function to get soft node assignments. Experiments were performed on a single GPU Tesla V100 with 32 GB of memory. Local solvers use as a loss function the sum of the Hamiltonian, see Sec. 2, and the Mean Squared Error (MSE) between the solution at time $t$ and the AM solution. We tested several alternatives, but MSE had better consistency across graphs. For the

*Main* GNN solver, the loss function is solely the Hamiltonian using the original QUBO matrix (per problem). Both solvers run until convergence, defined using a tolerance parameter or a maximum number of epochs. We set the maximum epochs to 10k and 1k for the *Main* and *Local*, respectively.

**Annealing Machine** We used a CMOS-based AM that implements momentum annealing [23]. Unlike the standard simulated annealing that updates variables individually, momentum annealing updates all connected variables simultaneously. Momentum annealing executed on a GPU is much faster than the simulated annealing performed on a CPU. We used an NVIDIA Tesla V100 GPU to perform the momentum annealing. Our AM can handle up to 100k decision variables. Variables were updated 1k times during each run of the annealing process. The momentum annealing was performed 1k times, and the best solution was retrieved.

**Graph Solver Variants** We compared three solver variants: *i) Raw GNN* (rGNN), a single GNN that takes as input the original graph, i.e., *Main* (global) solver; *ii) Multiresolution GNN* (mrGNN), a GNN-based local solver that receives the compressed versions of the original graph and solves the local QUBO problem. After convergence, the resulting node vectors are pooled following the Louvain schema to initialize the node vectors of the main GNN solver; *iii)* mrGNN+AM, a mrGNN with local guidance from the AM.

## 5   Results and Discussion

**Solution Quality and Convergence** We sampled solutions 50 times per target graph and obtained the final solution. We experimented on large graphs; therefore, no ground truth is available. Given this limitation, we used the final loss value as a measure of the quality of the solution, assuming that, in the absence of violations, a lower value means a better solution. Let *loss* be the evaluation of the Hamiltonian on a binarized solution; see Sec. 2. Unlike the relaxed version we used during training, this loss is our original optimization objective. We also checked violations based on the problem definition. In this scenario, a good solver achieves the lowest loss and, simultaneously, the minimum number of violations. For MIS, rGNN is faster than the other alternatives, reaching loss values 48% larger with 15% more violations on average. This pattern persists across problems, providing evidence that a single GNN block trained in a purely unsupervised way, while fast, seems unable to provide high-quality results. Between mrGNN and mrGNN +AM, while mrGNN is consistently faster, it also produces more violations across graphs, mainly for the largest graph. This behavior indicates that the information from the local GNN-based solver is indeed useful, compared to rGNN, but not enough to beat the contribution of an AM-based solver. We omit rGNN from the following analyses and focus on the trade-off between quality and speed of mrGNN and mrGNN +AM. Our comparison deals with *i)* a main GNN solver that received AM's information against and *ii)* a main GNN solver that received information only from the GNN-based solvers. Given the lack of ground truth (global minimum), we employed the *Relative loss difference*. It computes the difference between absolute values of the minimum loss achieved by mrGNN and mrGNN +AM models: $\Delta_{\mathrm{rel}} = \frac{|\mathrm{loss}_{\mathrm{mrGNN+AM}}| - |\mathrm{loss}_{\mathrm{mrGNN}}|}{|\mathrm{loss}_{\mathrm{mrGNN}}|} \times 100$. $\Delta_{\mathrm{rel}} > 0$ means mrGNN+AM has a lower loss than mrGNN. Table 4 presents the values of $\Delta_{\mathrm{rel}}$ across all target graphs for the three selected problems. In addition, Table 2 provides insights into how well models handle constraints. For MIS, we present the number of violations and how balanced the resulting sets are in the case of GP (ratio of their number of nodes).

Our approach is particularly effective for larger graphs. For smaller graphs (up to 100k nodes), mrGNN performs comparatively well. This performance reinforces our initial goal of using the GNN as a bridge to bring the accurate problem-solving capabilities of the AM to large-scale graphs. The quality of mrGNN degrades as we expand to more complex graphs (in terms of $n$ and $d$), where violations increase compared to mrGNN+AM. Table 4 shows the relative differences in terms of total execution time, $\Delta_T = \frac{\mathrm{time}_{\mathrm{mrGNN+AM}} - \mathrm{time}_{\mathrm{mrGNN}}}{\mathrm{time}_{\mathrm{mrGNN}}} \times 100$. $\Delta T > 0$ means mrGNN is faster than mrGNN+AM. Interestingly, if we look at the execution time comparison, there is a considerable difference depending on the combinatorial problem. This difference is evident when comparing MIS, where mrGNN is much faster. We hypothesize that the primary input discrepancy resides in the QUBO matrix's shape, as the underlying graph structures remain the same. Therefore, differences may be due to solver technicalities, such as the sparsity[2]. Note that, for a given graph, a local

---

[2]In fact, preliminary experiments showed that training time could be further optimized by using a sparse representation of the QUBO matrix.

GNN-based solver is, on average, $9\times$ faster than the local AM-based solver, ignoring solution quality aspects. This speed advantage of the GNN solver underscores its practicality in real-world scenarios. We report the total time: the sum of local AM and GNN global solver times; the price to pay for a better solution is the extra time the AM takes. It is worth noting that such time comparison assumes the Louvain compression has been performed in advance, a realistic scenario in the real world.

**Impact of GNN module selection** We compared GCN against a Graph Attention Network (GAT) [29, 4], as the latter automatically learns to weight incoming node vectors from the set of neighbors during the aggregation step. For MIS, we could see a slight decrease in the achieved loss by up to 100k nodes, but at the expense of $8\times$ total time (average for all target graphs). Unfortunately for the other problems, no conclusive evidence was found across graphs, which may suggest the GNN layer could be problem-specific.

| Last Louvain used | $n$ | $\Delta$ MIS | $\Delta$ MaxCut |
|---|---|---|---|
| 1st | 67 146 | 5.94% | 3.10% |
| 2nd | 30 935 | 3.84% | 1.74% |
| 3rd | 14 641 | 0.11% | 1.02% |
| 4th | 7 013 | 0.01% | 0.21% |

Table 1: Difference in terms of final loss value using the $k$-th Louvain compressed graphs against using all of them for MIS and MaxCut on ($n$=150k, $d$=5) graph.

| | n | $d = 3$ | $d = 4$ | $d = 5$ |
|---|---|---|---|---|
| MIS violations | 50k | 0 / 0 | 0 / 1 | 0 / 3 |
| | 100k | 0 / 1 | 0 / 1 | 0 / 2 |
| | 150k | 1 /1 | 2 / 2 | 1 / 2 |
| GP ratio balance | 50k | 0.48 / 0.46 | 0.46 / 0.46 | 0.44 / 0.44 |
| | 100k | 0.46 / 0.43 | 0.44 / 0.44 | 0.46 / 0.43 |
| | 150k | 0.42 / 0.39 | 0.41 / 0.40 | 0.39 / 0.39 |

Table 2: Performance of AM-based and GNN-based methods. Ratio denotes AM /GNN.

**Node Decision Assignment Uncertainty** Fig. 2 shows the decision assignments to nodes during training. While, in general, there is progressive accumulation at the extremes, in red, we can see some cases that exhibit shifts. This phenomenon is undesirable as it could hinder early stopping. We focus on characterizing *late* shifts, i.e., changes at the late state of training. To do that, we uniformly divided the total number of epochs into three segments: *early*, *mid*, and *late* and counted how many shifts occur in each segment for both mrGNN and mrGNN+AM. Table 3 shows the proportion of shifts per segment for all target problems. This example shows a clear difference in the proportion of late-stage shifts, with mrGNN+AM reducing them in most cases. This behavior is consistent across graphs, which are omitted due to space limitations.

| Problem | Solver | early | mid | late |
|---|---|---|---|---|
| MIS | mrGNN | 0.46 | 0.19 | 0.35 |
| | mrGNN+AM | 0.53 | 0.25 | **0.22** |
| MaxCut | mrGNN | 0.68 | 0.21 | 0.1 |
| | mrGNN+AM | 0.77 | 0.21 | **0.03** |
| GP | mrGNN | 0.44 | 0.31 | 0.25 |
| | mrGNN+AM | 0.54 | 0.27 | **0.19** |

Table 3: Proportion of assignment shifts per training stage for various problems on the $n = 150$k, $d = 5$ graph. mrGNN+AM solver consistently produces less late stage shifts.

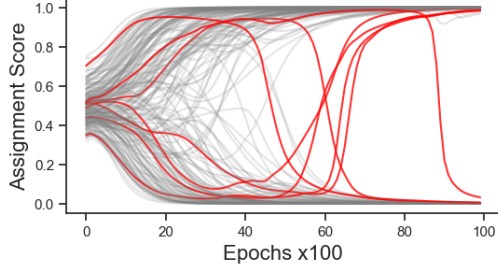

Figure 2: Evolution of the assignment scores from for a sample of 200 nodes for the MaxCut problem on the $n = 50$k, $d = 3$ graph using mrGNN.

**Comparison Against other Baselines** For ML-based solvers, most works operate only on small to medium-size graph benchmarks, such as [5], which represents the state-of-the-art for MIS, considers benchmarks less than a hundred nodes (in average). The assumption is the same for classical (heuristic) solvers. While they can be very efficient on small to medium-sized graphs, they cannot provide comparable solutions as size increases. We conducted the same experimental setting for graphs with sizes 50K, 100K, and 150K varying degrees in $[3, 4, 5]$ for the MIS problem to assess such an assumption. We selected two representative classical algorithms: Greedy Search (GS) and the HB algorithm [3]. Results confirm our central hypothesis on the difficulty of scaling up classical heuristics: GS finished within 2h only for the family of graphs with $n$=50K, with an MIS of size 17% lower than the one obtained by the proposed model. For larger graphs, we stopped the execution after 2h, considering that the total time for the proposed approach was, on average, 12 min. HB algorithm did not finish for $n$ =50K. In light of this evidence, we consider our approach to be able to scale and

|      | n    | $\Delta_{rel}$ | | | $\Delta T$ | | |
|------|------|-------|-------|--------|--------|--------|--------|
|      |      | $d=3$ | $d=4$ | $d=5$ | $d=3$ | $d=4$ | $d=5$ |
| MIS | 50k | -4.83% | -5.77% | -5.556% | -79.12% | -66.31% | -46.6% |
|     | 100k | 25.06% | 28.11% | 33.248% | 21.04% | 21.86% | 24.7% |
|     | 150k | 25.91% | 31.07% | 34.731% | 63.11% | 69.01% | 89.4% |
| MaxCut | 50K | 0.22% | 0.096% | 0.031% | 3.57% | 1.07% | -9.61% |
|        | 100k | -0.755% | 0.351% | 0.156% | 1.94% | -7.601% | -11.27% |
|        | 150k | 9.001% | 8.767% | 8.94% | -13.06% | -19.44% | -19.83% |
| GP | 50K | 0.098% | 0.111% | 0.141% | 9.57% | 11.04% | 11.37% |
|    | 100k | -1.43% | 2.095% | 2.641% | 3.11% | -1.99% | -4.08% |
|    | 150k | 2.063% | 1.812% | -1.773% | -1.67% | 0.57% | -3.99% |

Table 4: Relative time and loss differences between mrGNN+AM and mrGNN

provide suitable solutions where classical methods cannot. Finally, we considered a *neural* model (NM), as presented in [27]. In this case, we compare at MIS size obtained by MN and our proposed method. Out of the nine target graphs, NM only obtains higher MIS size for three: 50K, d= [3,4] and 100K, d=3. For the rest, our approach obtains on average of 4.9% higher MIS values, at equal or less number of violations.

## 6 Conclusion and Future Work

We explore how to combine the accuracy of AM and the flexibility of GNN to solve combinatorial optimization problems. Our approach was tested on canonical combinatorial problems, showing that the flexibility of GNNs can allow the transfer of the accurate capabilities of the AM to graphs that are initially out of its reach. For future work, we are interested in *i)* the reuse of partial solutions across similar problems and *ii)* an end-to-end differentiable framework.

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
