# OpenReview forum: "Annealing Machine-assisted Learning of Graph Neural Network for Combinatorial Optimization"
_NeurIPS.cc/2024/Workshop/MLNCP — MLNCP Poster_

### Official Review · Reviewer_TTMr · 2024-09-19

**Rating:** 6
**Confidence:** 3

**Review:**

In this paper, the authors propose a way to combine graph neural networks (GNNs) with annealing machines (AMs) to solve large-scale combinatoric optimization problems. The idea is to compress the large graph into small chunks, execute annealing machines on the small chunks, and aggregate the resulting features as reference information for a main graph neural network solver. This enables solving problems that are too large to be solved with AM, while providing more accurate results than GNN thanks to the information given by AM. The authors conduct numerical experiments to evaluate the performance and computation time of the proposed GNN+AM scheme, in comparison with GNN alone.

Strengths:

1. The idea of combining GNN and AM is interesting and seems to indeed enable solving large scale combinatoric optimization problems.

2. The authors considered many prototypical optimization problems, including maximum cut, maximum independent set, and graph partition. They conducted extensive numerical experiments to evaluate the performance of different GNN modules.

Weaknesses and comments:

1. I think the performance gain of combining GNN and AM is not very convincing. As the authors also point out, the performance on maximum cut and graph partition does not show a consistent trend and seems model-specific. I understand that this is only preliminary results. But the authors should consider doing more experiments to analyze why this is the case.

2. The presentation of Algorithm 1 should be self-contained. In particular, every newly introduced variable should be given a way to calculate. For example, the calculation of $\bar{R}^i$ should be stated explicitly.

---

### Official Review · Reviewer_XVo5 · 2024-09-26
**This novel approach attempts to combine annealing machines and graph neural networks to solve complex combinatorial problems.  Overall, this preliminary work is well-described and well-written.**

**Rating:** 7
**Confidence:** 3

**Review:**

This work seeks to combine the advantages of using graph neural network (GNN) approaches, which are they are generally faster and scalable, with annealing machine (AM) approaches, which are generally more accurate but don't scale as well, on complex combinatorial optimization problems.  In general, I find the concept of this work compelling, and the preliminary results are interesting.  If they are able to effectively combine these two approaches, it can help address issues with scalability of of the AM approaches while also improving GNN performance.

Pros:

-- Novel approach to improve combinatorial optimization

-- Well-written and well-described

-- Good comparison with other approaches as well

-- Appreciated the inclusion of both accuracy and time as metrics

Cons:

-- I would like to have seen results on real-world graphs

-- I would like to have seen more discussion about where the GNN+AM approaches are worse

---

### Decision · Program_Chairs · 2024-10-10

Accept (Poster)